# Complementary Protection in Japan: To What Extent Does Japan Offer Effective International Protection for Those Who Fall Outside the 1951 Refugee Convention?

**Brian Aycock** [1,*] and **Naoko Hashimoto** [2]

1   Graduate School of Arts and Sciences, International Christian University, Tokyo 181-8585, Japan
2   Graduate School of Social Sciences, Hitotsubashi University, Tokyo 186-8601, Japan; naokoh@kg8.so-net.ne.jp
*   Correspondence: aycock.brian@gmail.com

**Abstract:** This study focuses on what Japan's Immigration Control and Refugee Recognition Act (ICRRA) calls 'Special Permission to Stay' (zairyū tokubetsu kyoka) on humanitarian grounds (SPS), and evaluates the extent to which SPS provides effective international protection for those who are not recognized as refugees in Japan. The evaluation uses the European Union's Qualification Directive (QD) as a yardstick. This paper explains the legal framework through which Japan offers complementary protection and explores the application of the law in practice. By investigating cases of SPS granted in Japan over a five-year period, the authors infer the prevailing legal interpretations on critical elements of complementary protection policy not clearly defined in the ICRRA. Case law is not widely available in Japan, but the authors have analysed all of the available case summaries provided by the Ministry of Justice only in Japanese. This work represents the first research conducted in English into these summaries. Further, several elite interviews were conducted with key senior immigration officials to gain insight into the inner workings of the Japanese system of SPS. Based upon the empirical evidence collected, the research demonstrates that the ICRRA often lacks clarity and is too discretionary, but that it also provides flexibility that allows a more inclusive application of the law.

**Keywords:** subsidiary protection; complementary protection; refugee law; EU; Japan





## 1. Introduction

The definition of refugees enshrined in Article 1A(2) of the 1951 Convention Relating to the Status of Refugees (hereinafter 'the Refugee Convention') is narrow and restrictive. Despite the evolving interpretations of such key concepts as persecution and the membership of a particular social group, those facing serious harm from war and generalised violence, for instance, are normally viewed as falling outside the refugee definition (UNHCR 1979, paras 164 through 166). Alternative forms of protection are often required for persons not qualifying under the restrictive requirements of the 1951 Convention, and many States and regions have devised alternative protection schemes to include vulnerable persons who are not recognized as refugees. One example is found in Australia's prohibition against refoulement, enshrined in the Migration Act of 1958 (Migration Act 1958, Act No. 62). Another form is found in the United States' Temporary Protection Status, often used as a form of protection in the face of short-term crises, like natural disasters or armed conflicts (Immigration and Nationality Act 1952). By far the most substantial effort to codify and standardize a system of complementary protection is that of the European Union's recast Qualification Directive (QD). Leading scholars in the field maintain that '[s]ubsidiary protection encompasses categories that go beyond the refugee definition' (Tsourdi 2014, p. 272), and that the 'growth of complementary forms of protection—as a way of reflecting international human rights obligations—is notable' (Harvey 2015, p. 46).

Although little attention has been hitherto paid, Japan is no exception to this global trend. Since 1991, a much larger number of foreign nationals has been provided a complementary form of protection than those granted an official refugee status in Japan, as will be discussed below. More specifically, this study focuses on what Japan's Immigration Control and Refugee Recognition Act (ICRRA 2009) (ICRRA) calls 'Special Permission to Stay' (zairyū tokubetsu kyoka) (ICRRA 2009, Art. 61-2-2(2)) on humanitarian grounds (hereinafter referred to as SPS), and evaluates the extent to which SPS provides effective international protection for those who are not recognized as refugees in Japan. Given the current international climate, it is not realistic to expect a significant overhaul of the 1951 Convention in a manner to allow for a broader inclusion than the current refugee definition. Thus, complementary protection will remain an important mechanism for protecting those who are at risk of serious harm but who do not qualify as refugees. This is especially true in countries like Japan where the State and the courts insist on a narrow and restrictive interpretation of the 1951 Convention refugee definition. Japan has, historically, provided protection through SPS more often than it has recognised asylum seekers as refugees, and that makes this aspect of Japanese law worthy of close scrutiny. It could be argued that SPS is the primary form of international protection offered in Japan.

In evaluating Japan's SPS, one requires a certain set of standards and yardsticks. There is no singular definition for complementary protection in the world, and there is no instrument in international law beyond the regional level that directly addresses it. It is therefore inevitable to examine efforts at another national or regional level. This paper employs subsidiary protection found in the European Union's Qualification Directive (QD 2011) (QD) as a yardstick, for reasons provided at the end of the methodology section below. Meanwhile, it is important to emphasize at the outset that one cannot directly compare SPS with the more robust legal framework for subsidiary protection. The QD is a part of the broader Common European Asylum System (CEAS), and it is further expanded and nuanced by mechanisms and instruments at various national levels within the EU. Further complexities would emanate from the differences between continental law and common law traditions. Therefore, this paper offers an evaluation of Japan's SPS using the EU's QD as a yardstick, without attempting a direct comparison, to highlight how Japan protects those falling outside the 1951 Convention but still requiring international protection.

Concomitant with the actual developments of subsidiary or complementary forms of protection, academic scholarship related to the subject has also steadily been enriched. Some notable works include those by Zimmermann and Wennholz (Zimmermann and Wennholz 2001), Goodwin-Gill (Goodwin-Gill 1986), Perluss and Hartman (Perluss and Hartman 1986), Weiss (Weis 1978), McAdam (McAdam 2005a), Hailbronner (Hailbronner 1986), Lauterpacht and Bethlehem (Lauterpacht and Bethlehem 2003), and others. They provide insights on international standards and practice, as well as the legal basis of State obligations. In particular, McAdam's works related to subsidiary protection generally (McAdam 2007), and on the QD specifically (McAdam 2005b), have been influential in this research. Other significant contributions to the understanding of the QD are found in works by Storey (Storey 2008) and Gil-Bazo (Gil-Bazo 2006). However, none of these addresses refugee law in Japan or SPS in particular. It is hoped that this paper will, in some small measure, add to this existing body of scholarship by providing insights into Japan's immigration law and, also, by contributing to the growing discussion on international protections for those falling outside the Refugee Convention definition.

It is true that there are also a number of works related specifically to Japan, its legal system, and its role in the international refugee regime. Yuji Iwasawa's book, International Law, Human Rights, and Japanese Law (Iwasawa 1998), provides the framework for understanding the relationship between international law (e.g., the Refugee Convention) and domestic Japanese law (e.g., ICRRA). Osamu Arakaki's work offers one of the most comprehensive overviews of the refugee status determination procedure in Japan, as well as very useful criticisms of the system (Arakaki 2016). Additionally, the work of Yukari Ando in highlighting Japan's narrow interpretation of persecution, which has significant

impacts on the refugee determination procedure (Ando 2016). Particularly important to this research project is the previous work of Naoko Hashimoto, on the stratification of rights among forced migrants in Japan (Hashimoto 2019). Her detailed analysis informs this article with much of the information needed to evaluate the content of protection provided by the Japanese system of SPS against the EU's QD. Again, while these works significantly inform this article, none of them directly address SPS.

Against these backdrops, this article aims to achieve two objectives which are often portrayed as the 'dual imperative' (Jacobsen and Landau 2003) in forced migration studies: contributing to the academic field of study and advocating for policy improvements on behalf of those seeking protection.

The first of these imperatives is met in the article's attempt to address the lacuna of academic attention to the legal status of applicants facing risks but not recognised as refugees in Japan. Japan acceded to the Refugee Convention (Refugee Convention 1951) in 1981 and, between January 1982 and December 2019, it recognised 794 individuals as a Convention refugee out of a total of 81,543 asylum seekers in Japan. This brings the total refugee recognition rate to less than one percent during the four decades. The unusually high rate at which refugee claims are rejected in Japan is striking, and the reasons for this have been analysed elsewhere (Arakaki 2016) (Ando 2016) (Hashimoto 2018). In parallel to the full-fledged refugee recognition, the Japanese Government has also granted SPS for some asylum seekers since 1991. By December 2019, a total of 2665 individuals benefited from this status. At least numerically, SPS has provided a larger number of asylum seekers with a certain form of protection in Japan than has the formal asylum system under the 1951 Refugee Convention. Nevertheless, this potentially more significant form of protection has rarely attracted any robust academic scrutiny from scholars. Even with the increasing international attention to complementary protection in international refugee law over the last two decades, the case of Japan has remained as an academic vacuum in international refugee law scholarship as mentioned above. One of the contributions this article attempts to make is to fill this gap.

The other contribution is praxis. In 2013, a Task Force on Refugee Status Determination (RSD) Procedure consisting of independent experts was established by the Immigration Bureau of the Japanese Ministry of Justice. In its final report issued in December 2014, the Task Force recommended, inter alia, the codification of subsidiary protection in specific reference to EU's Qualification Directive (Task Force on Refugee Status Determination Procedure 2014). This recommendation was finally taken up in 2020, and the Immigration Services Agency (The Immigration Bureau was expanded and upgraded to the Immigration Services Agency as of 1 April 2019) drafted a bill to reform the ICRRA, although the formal submission of the bill to the Japanese Diet was postponed until 2021. Given the ongoing momentum within Japan's Immigration Services Agency to reform Japan's SPS and create a new system closer to EU's subsidiary protection, robust scrutiny of the two systems is as timely as ever, as it could provide practical inputs to the reform process. It is hoped this short piece of work achieves, in some small measure, both of these academic and practical goals.

The remainder of this paper is organized as follows. Section 2 explains the methods employed to collect data and empirical evidence and justifications to use the QD as a yardstick for this paper. Section 3 provides historical background to and evolution of SPS, based primarily upon direct elite interviews with Japanese decision-makers. Section 4 engages with robust and critical scrutiny of SPS by thematically grouping various articles and elements of QD and SPS into four general areas: eligibility, exclusion, content of protection, and procedure. The evaluation includes, inter alia, how each instrument defines relevant terms, what is required or prohibited or allowed, and what is included or excluded. The conclusion section discusses the strengths and weaknesses of SPS in light of the QD and provides a final answer to the research question as regards the extent to which Japan's SPS provides effective international protection for those whose refugee status is not recognized in Japan. The final section emphasises the need for developing Japanese law for providing

subsidiary protection for forced migrants, as well as the need for further research. The paper concludes with summarising key strengths inherent to SPS, particularly its potential for inclusivity, while also recognising weaknesses, most notably its lack of specificity.

## 2. Methodology

This study primarily employs three methods in collecting relevant data and empirical evidence. First, the research was conducted through qualitative legal analysis on existing legislation and case law. This primarily focuses on Japan's ICRRA, the EU's QD, the 1951 Convention, and the related jurisprudence. Jurisprudence from both the EU and Japan has been used to show how various legal instruments (see Appendix A) have been interpreted, and to serve as examples of the application of law in practice. It should be noted that the available case law from Japan is limited. Japan's Immigration Services Agency, a department of the Ministry of Justice, provides statistics on how many refugees have been admitted and how many cases have been granted SPS. The Ministry also provides brief summaries of a limited number of cases. This project conducts an exhaustive study of all of the available cases, including the summaries provided by the Ministry of Justice: 12 cases in 2015; 15 cases in 2016; 6 cases in 2017; 7 cases in 2018; 5 cases in 2019. Summaries on refugee status determinations have only been provided since 2012 (and no SPS case summaries were provided between 2012 and 2014) and only in Japanese, creating a significant barrier to research from foreign scholars. This research contributes to scholarship for the first time in the field by providing information not otherwise available in English.

Figure 1 shows how many people have been offered protection, through third-country resettlement, refugee recognition of asylum seekers applying within Japan, and SPS, as well as how many summaries are available from the cases granted SPS between 2015 and 2019. It demonstrates that the case summaries published provide only a partial picture of the entire SPS cases. There is no guarantee that the published summaries are representative of other cases, and no way of knowing how the Immigration Services Agency decides which cases to publish or the reasoning. More importantly, the available case summaries do not provide any information concerning the grounds on which the SPS status is denied, as we will discuss in more details in the subsequent section. While duly acknowledging these data limitations, the authors use the publicly available case summaries to infer deductively decision-making thresholds and criteria.

Secondly, the authors examined published scholarly works in the relevant field, i.e., 'the teachings of the most highly qualified publicists of the various nations' (ICJ 1946, Art. 38) and grey literature. Authoritative interpretations and legal analysis have been put forward by a number of leading scholars, as already enumerated in the introductory section above, particularly as pertains to EU's subsidiary protection. In the absence of definitive global standards in providing complementary protection, this study incorporates such authoritative legal analysis when elucidating and clarifying the yardstick to be extracted from EU's subsidiary protection against which to evaluate Japan's SPS. When it comes to Japan's SPS, several retired Japanese immigration officials published personal memoires and legal commentaries in relation to Japan's refugee protection policy and ICRRA. The most notable and useful examples of such grey literature for the purpose of this study include: Susumu Yamagami's *Gekihen no Jidai: Wagakuni to Nanmin Mondai: kinou, kyou, ashita* (*The turmoil era: my country and refugee issues: yesterday, today and tomorrow*) (Yamagami 2007) and Kazuteru Tagaya and Shigeru Takaya's comprehensive commentaries on each article of ICRRA, entitled *Nyukanhou Taizen* (Tagaya and Takaya 2015). This study is the first of its kind in fully incorporating the perspectives and interpretations published by retired senior immigration officials in academic scrutiny of Japan's SPS.

The third method involved elite interviews with Japanese immigration officials. The author conducted semi-structured interviews with four senior Japanese immigration officials. Susumu Yamagami worked for the Japanese Immigration Bureau for 30 years (from 1972 to 2002) and handled Japan's accession procedure to the Refugee Convention as the main officer in charge; Shigeru Takaya served from 1981 to 2013 and retired as the

Director-General of the Immigration Bureau; and the other two are incumbent senior immigration officers. A total of six interviews were conducted via either online zoom meetings or private email, each interview session lasting for about one and a half hours, between June and October 2020 in Tokyo, Japan. The authors obtained informed consents from Yamagami and Takaya to quote their statements, names, and designations in this paper, while the two incumbent officials preferred to remain anonymous while permitting their statements to be quoted. Research ethics approval, including engaging human participants, was received through the University of London's Refugee Law Initiative.

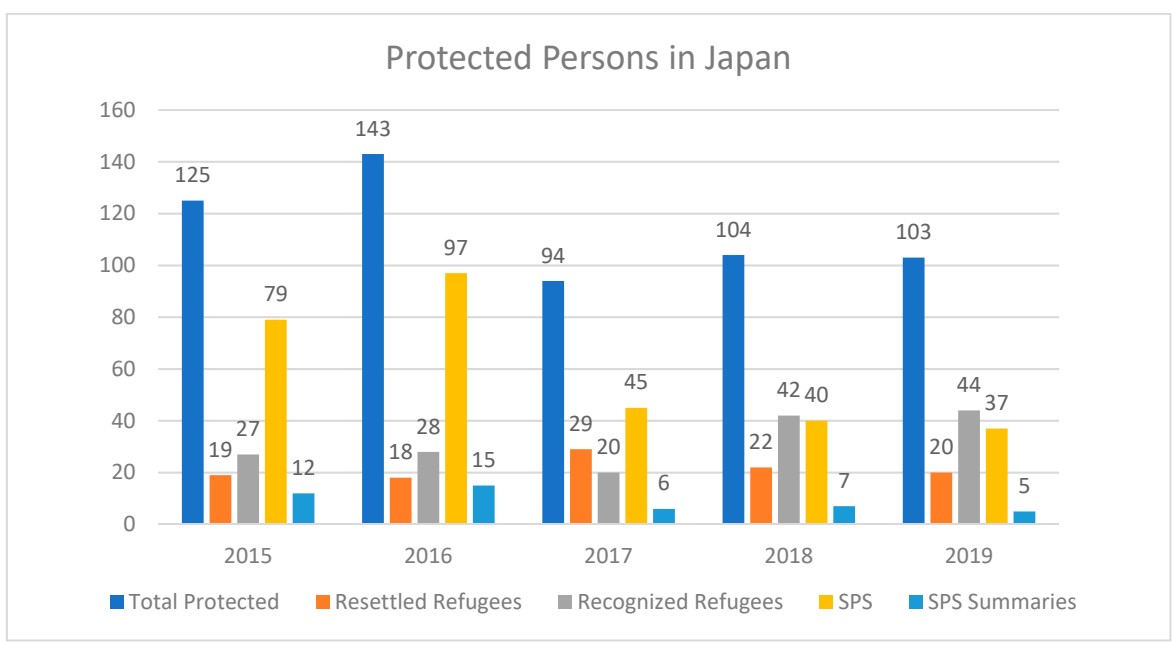

**Figure 1.** Protected persons in Japan. (Source: Immigration Bureau, Ministry of Justice (Japan), 2018 Report on the Status of Refugee Protection in Japan (Wagakuni ni okeru nanmin higo no jōkyō-tō), http://www.moj.go.jp/content/001290415.pdf [accessed 12 October 2019].

Based on the data and empirical evidence collected by these three methods, this article attempts to evaluate Japan's SPS vis à vis subsidiary protection provided under EU's QD. There are three major reasons why such evaluation is justifiable. The first reason, as already mentioned above, is that the final report of the Task Force in 2014 recommended the creation of a new subsidiary protection system in Japan specifically in reference to EU's QD.

Second, there is no internationally agreed-upon principle regulating how, when, and to what extent international protection should be provided for those foreigners who fall outside Art. 1A(2) of the 1951 Refugee Convention. Various governments, either unilaterally or multilaterally, have devised agreements not to refoul some foreigners who may face danger and/or human rights violation upon return to their country of origin, one of the examples for which is EU's QD. In light of Art. 31-3 (general rule of interpretation) of the 1969 Vienna Convention on the Law of Treaties (hereinafter the 1969 Vienna Convention), EU's QD could be regarded as 'any subsequent agreement between the parties regarding the interpretation of the treaty or the application of its provisions' which a state party to the Vienna Convention shall take into account when interpreting an international treaty—in this case the 1951 Refugee Convention. In other words, Japan is to at least take into account—if not be bound by—EU's QD when interpreting the 1951 Refugee Convention by virtue of the fact that Japan, as well as all EU member states, is party both to the 1951 Refugee Convention and the 1969 Vienna Convention. Significantly, Gil-Bazo noted that the QD 'constitutes the first supranational legally binding instrument in Europe that recognises the status of individuals protected under international human

rights law' (Gil-Bazo 2006, p. 14), which makes it both the most substantial legal instrument against which to evaluate Japan's SPS and the kind of protection it offers, and a highly influential instrument in light of the 1969 Vienna Convention obligations. While other States offer domestic examples against which one could measure Japan's SPS, the facts that the Japanese government has specifically noted the EU QD as an example and that it offers the most highly developed example, and in consideration of Art. 31-3 of the 1969 Vienna Convention, all support our view that the EU QD is the most appropriate benchmark to employ in this study.

The third reason is that Japan currently has either the full-fledged asylum in accordance with the 1951 Refugee Convention or the SPS system, with no subsidiary protection per se. SPS has been thus serving both as a residential status for failed asylum-seekers who have various humanitarian reasons to stay in Japan, and as a de facto alternative to EU's subsidiary protection, as will be analysed in the subsequent sections. In light of this, some might contend that Japan's SPS should be more comparable to 'humanitarian admission status' devised under various national legislations in other countries. Meanwhile, as long as SPS covers some of the functions played by subsidiary protection under EU's QD, the authors believe that Japan's SPS should be evaluated using EU's subsidiary protection as a yardstick, until a robust, full-fledged subsidiary protection system is newly established in Japan, hopefully in the near future.

### 3. Historical Background and Evolution of SPS

Table 1 provides comprehensive asylum statistics of Japan from 1978 (when the first Indo-Chinese refugees were allowed to reside in Japan) until 2019. Although the Indo-Chinese refugee admission program still accounts for by far the largest part of Japan's entire asylum history, SPS functioned as the major route to access protection in Japan between 2005 and 2017, rather than the Convention status.

It was 1991 when the first cases of asylum seekers were provided SPS on humanitarian grounds under Article 50 of the initial/old Immigration Control and Refugee Recognition Act (ICRRA) of Japan. ICRRA was promulgated in 1982 on the occasion of Japan's accession to the Refugee Convention, and Art. 50 was created as a general provision to grant a special permission for residence regardless of whether or not the foreigner was seeking asylum in Japan (Hashimoto 2019, p. 129). According to Susumu Yamagami, a retired Immigration Bureau official who directly handled Japan's accession procedure to the Refugee Convention in the early 1980s and who devised this innovative measure at that time, the idea was based upon the experience of protecting Indo-Chinese refugees since the 1970s. Since around 1985, the Immigration Bureau had started discussing the need to provide some asylum seekers with a permission to stay in Japan on humanitarian grounds even if they could not be recognised as a Convention refugee.

Yamagami stated in his email reply to the author, as follows:

> '*Since the Indo-Chinese era, we were aware that some of the displaced people would not meet the refugee definition enshrined in the Refugee Convention. The question then was whether we should just recognise them as a refugee purely based upon their claim, or we would deport them if there was no evidence to back up their claim. My opinion was always that there would be some people whom we should allow to stay in Japan even if we could not recognise them as a refugee. I was also aware that some European countries granted such claimants so-called "B status" . . . '* (For more information on 'a humanitarian status' or 'Status B', see UNHCR 1989) (a personal email from Susumu Yamagami to the author dated 13 July 2020)

As a result, between 1991 and 2004, a total of 284 individuals were granted the SPS under Art. 50 of ICRRA.

**Table 1.** Japan's refugee protection statistics 1978–2019.

| Year | Asylum Applicants | Refugees Recognised | Indo-Chinese/ Resettlement | Special Permission to Stay |
|------|------|------|------|------|
| 1978 | - | - | 3 | - |
| 1979 | - | - | 94 | - |
| 1980 | - | - | 396 | - |
| 1981 | - | - | 1203 | - |
| 1982 | 530 | 67 | 456 | - |
| 1983 | 44 | 63 | 675 | - |
| 1984 | 62 | 31 | 979 | - |
| 1985 | 29 | 10 | 730 | - |
| 1986 | 54 | 3 | 306 | - |
| 1987 | 48 | 6 | 579 | - |
| 1988 | 47 | 12 | 500 | - |
| 1989 | 50 | 2 | 461 | - |
| 1990 | 32 | 2 | 734 | - |
| **1991** | 42 | 1 | 780 | 7 |
| 1992 | 68 | 3 | 792 | 2 |
| 1993 | 50 | 6 | 558 | 3 |
| 1994 | 73 | 1 | 456 | 9 |
| 1995 | 52 | 2 | 231 | 3 |
| 1996 | 147 | 1 | 151 | 3 |
| 1997 | 242 | 1 | 157 | 3 |
| 1998 | 133 | 16 | 132 | 42 |
| 1999 | 260 | 16 | 158 | 44 |
| 2000 | 216 | 22 | 135 | 36 |
| 2001 | 353 | 26 | 131 | 67 |
| 2002 | 250 | 14 | 144 | 40 |
| 2003 | 336 | 10 | 146 | 16 |
| 2004 | 426 | 15 | 144 | 9 |
| 2005 | 384 | 46 | 88 | 97 |
| 2006 | 954 | 34 | - | 53 |
| 2007 | 816 | 41 | - | 88 |
| 2008 | 1599 | 57 | - | 360 |
| 2009 | 1388 | 30 | - | 501 |
| 2010 | 1202 | 39 | 27 | 363 |
| 2011 | 1867 | 21 | 18 | 248 |
| 2012 | 2545 | 18 | 0 | 112 |
| 2013 | 3260 | 6 | 18 | 151 |
| 2014 | 5000 | 11 | 23 | 110 |
| 2015 | 7586 | 27 | 19 | 79 |
| 2016 | 10,901 | 28 | 18 | 97 |
| 2017 | 19,629 | 20 | 29 | 45 |
| 2018 | 10,493 | 42 | 22 | 40 |
| 2019 | 10,375 | 44 | 20 | 37 |
| Total | 81,543 | 794 | 11,513 | 2665 |

Source: Immigration Service Agency of Japan 2020.

When the ICRRA went through major revisions in 2004, a new provision was created to regulate special permission for residence on humanitarian grounds specifically for asylum seekers who would not meet the Convention refugee criteria, i.e., Article 61-2-2 (ii). There were two rationales behind this revision. One was to regularise post facto practice of granting special permission of residence for asylum seekers not meeting the refugee definition, as mentioned above. The other is to streamline and combine the refugee status determination procedure on the one hand, and the question of granting of residential status on the other. In fact, up to 2004, even those officially recognised as a refugee were not automatically granted a residential status. Particularly unlawfully staying refugees had to go through the examination of special permission to stay under Art. 50 as an irregular migrant after the whole RSD procedure. This not only presented procedural burdens both for refugees and the Immigration officials but also, and more importantly, had a risk of leaving Convention refugees in a legal limbo without residential status in Japan. To streamline and combine the RSD and SPS procedures, the new Art. 61-2-2 was created and (ii) was added as a SPS specifically for asylum seekers not meeting the refugee definition, to make it separate from the special permission for residence for general migrants under

Art. 50 (Interviews and personal communications with Susumu Yamagami and Shigeru Takaya between June and October 2020).

However, even the new provision under Art. 61-2-2 (ii) was not without problems. The major focus of the 2004 revisions was on other issues including (a) the abolishment of the so-called 60 days rule, which required that asylum applications must be lodged within 60 days upon entry to Japan or from the time when the applicant learnt the persecution risk at home, (b) establishment of a status of provisional stay for asylum seekers under certain conditions, and (c) creation of the Refugee Examination Counselor system, rather than the creation of SPS for asylum seekers. As a result, SPS under Art. 61-2-2 (ii) started being implemented without detailed criteria or procedure set out as to how, when, and under what conditions SPS should be granted for asylum seekers. Shigeru Takaya, the former Director-General of the Immigration Bureau, articulated this lack of clarity surrounding SPS for asylum seekers in his interview. He stated that "the practice [of granting SPS] continued without its eligibility criteria being clarified [by the Immigration Bureau] as a policy matter . . . ". (Interview with Tagaya on 9 October 2020. See also Tagaya and Takaya 2015).

When looking at Table 1, the statistics demonstrate the huge surge of the cases granted SPS between 2008 and 2014 and one might suspect if there was any policy change in granting SPS. Asked if there were any changes, a senior immigration official responded negatively in her interview, saying that the surge in the SPS numbers were due to increase and decrease in the asylum applicants from Myanmar, particularly given the so-called Saffron Revolution in 2007, not due to any change in SPS eligibility criteria (Anonymous interview with a senior immigration official on 9 October 2020). This practice of granting SPS without clear criteria has turned out to be a mixed blessing, as the following sections of this paper demonstrate.

## 4. Evaluation of SPS in Light of QD

As noted, the QD is the most robust instrument addressing subsidiary protection. It is far more detailed than Japan's SPS. It would be inefficient to go through each article individually when there are not corresponding articles in the ICRRA. Therefore, the most manageable way to evaluate Japan's SPS in light of subsidiary protection under QD is to group the articles and elements under these two legal instruments thematically. In this way, the authors are able to discuss topics that can be inferred from the ICRRA even though they are not specifically enumerated. As an overview, the elements involved include:

(1) Eligibility, or the granting of complementary/subsidiary protection
(2) Exclusion, cessation, and revocation of protection
(3) Content of protection
(4) Procedural issues

The following sections first identify standards provided in the QD, interpret them based on academic legal analysis, and evaluate Japan's SPS against them, for each of the four thematic groups.

### 4.1. Granting Complementary/Subsidiary Protection

4.1.1. The Requirement to Grant Protection

Thematically, subsidiary protection in the QD begins with Article 18. Member States are required to grant subsidiary protection to persons eligible under the rules defined in the QD, according to Art. 18. The importance of Art. 18 is that it requires, rather than allows, subsidiary protection. Significantly, there is no provision in the ICRRA requiring the granting of SPS. Where Art. 18 of the QD uses the phrase 'Member States *shall* grant subsidiary protection' (QD, Art. 18) (emphasis added), the ICRRA states that the 'Minister of Justice *shall* examine' (emphasis added) and '*may* grant' (ICRRA, Art. 61-2-2(2)) (emphasis added) SPS. This important distinction means that the QD has enumerated and codified the grounds on which a person becomes eligible for subsidiary protection and requires Member States to grant it. Japan, on the other hand, fails to enumerate any specific eligibility requirements, and it is left to the discretion of the Minister of Justice.

'Shall' signifies a mandatory requirement, or an imperative, where 'may' signifies the act is optional (Storey 2008, p. 17). Accordingly, the Minister of Justice in Japan is only required to examine whether there are grounds for granting SPS, but those grounds are not defined, as the retired senior immigration officials reconfirmed in their interviews with the author. It is arguably possible to contend that 'may', in this context, invokes an imperative if there are grounds, but that is impossible to establish if no grounds have been enumerated.

The vagueness of the ICRRA and the significant discretionary powers left in the hands of immigration officials are problematic. First, there is no objective standard one must meet to establish eligibility for SPS. Second, there is very little chance for appeal as there is no language establishing SPS as a right owed to anyone. Thirdly, the language of the ICRRA, coupled with the lack of accountability in the refugee determination procedure, makes scrutiny from the outside very difficult. If there are no specific eligibility requirements, and decisions on granting SPS are discretionary, and the Minister of Justice is never actually required to grant it, then by what standards can the process be objectively evaluated to determine fairness and consistency?

### 4.1.2. Eligibility Criteria

A key to understanding eligibility, and other pertinent rules, are found in the terminology as defined in the QD. Art. 2 of the QD defines the key terms to be used throughout the Directive (QD, Art. 2). Importantly, Art. 2(a) and Art. 2(b) codify that international protection refers to both refugees and those entitled to subsidiary protection. When examining other parts of the QD, then, one can be mindful that any reference to international protection is equally applicable to those whose refugee status was not recognized, but who do qualify for subsidiary protection. For the purposes of this inquiry, the most significant definition is found in Art. 2(f), which defines an eligible person for subsidiary protection as one 'who does not qualify as a refugee but in respect of whom substantial grounds have been shown for believing that the person concerned, if returned to his or her country of origin, or in the case of a stateless person, to his or her country of former habitual residence, would face a real risk of suffering serious harm as defined in Art. 15, and to whom Art. 17(1) and (2) does not apply, and is unable, or, owing to such risk, unwilling to avail himself or herself of the protection of that country' (QD, Art. 7(f)). Art. 15 and Art. 17 will be discussed in a later section. The important point in this section is that Art. 2(f) codifies who is eligible for subsidiary protection: someone facing a real risk of serious harm.

By contrast, the ICRRA states simply that when one is not recognized as a refugee the Minister of Justice 'may grant special permission to stay if he/she finds such grounds' (ICRRA, Art. 62-2-2(2)). Just as the requirement to grant protection is omitted, so too is any definition of the 'grounds' on which the Minister may grant SPS. This is arguably the most legally problematic aspect of SPS. Japan has failed to codify any specific conditions that might qualify someone for humanitarian status. With the granting of SPS left entirely to bureaucratic discretion, presumably a task delegated to the Refugee Inspector, raises concerns about the consistent application of the law.

A further shortcoming of the ICRRA when evaluated against the standard of the QD is the failure to provide a means of referencing jointly refugees and those granted SPS, as the QD does with the term 'international protection' (QD, Art. 2(a)). As there is no overarching concept of those under international protection, other references to refugees in the ICRRA do not explicitly apply to those granted SPS. A few examples include revocation of status (ICRRA, Art. 61-2-8), special provisions on permanent residence (ICRRA, Art. 62-2-11), and refugee travel documents (ICRRA, Art. 61-2-12). By not explicitly including those granted SPS in articles related to these matters, their rights and responsibilities may differ significantly from those of refugees. It is important to reiterate that the lack of clarity is problematic. This makes access to these rights tenuous, at best. As shown in the section on procedural issues, below, Art. 4 of the QD offers a good example of how an overarching term like international protection can ensure that applicable rights and responsibilities be codified for issues that relate to both refugee status and subsidiary protection.

With Art. 18 mandating the granting of subsidiary protection to eligible persons, and Art. 2 defining who is eligible, one can dig deeper into eligibility in Art. 15. In reference to the element of the Art. 2 criterion of risk of serious harm, Art. 15 enumerates acts that meet the eligibility requirement. Very simply, serious harm is defined as facing the death penalty, torture or inhuman or degrading treatment, or threat to life by indiscriminate violence in conflict (QD, Art. 15). Goodwin-Gill and McAdam rightly point out that the concept of serious harm 'is not part of international law and was devised for the purposes' of the QD (Goodwin-Gill and McAdam 2007), and it has not been accepted by Japanese courts. Despite it not being part of international law, this concept provides a useful example against which to measure the granting of complementary protection in other instruments like the ICRRA.

Meanwhile, certain elements of the QD's specificity may create problems that the ICRRA avoids through its flexibility. One example of the ICRRA's flexibility is found in Case 4 of the Ministry of Justice summary report on applicants in 2018. Japan failed to recognize a woman as a refugee because it was determined that the risk she faced in the country of origin did not reach the level of persecution. The applicant was granted SPS on the grounds that women are discriminated against, she is unlikely to be able to obtain stable employment, and as a divorced woman she requires stable employment to support her three children (Ministry of Justice Summaries 2018, Case 4). It is arguable that another jurisdiction might recognize such an applicant as a Convention refugee, interpreting this level of discrimination as persecution and recognizing gender as the particular social group to establish nexus. That said, the applicant would not have been eligible for subsidiary protection under Art. 15 of the QD because neither discrimination nor economic hardship nor child welfare are included in the definition of serious harm. McAdam calls Art. 15 a 'political compromise' that is 'conservative in its scope', emphasizing that it includes only 'the least contestable human rights-based protections' (McAdam 2005b, p. 474). This seems evident in the enumeration of only three general grounds for subsidiary protection: the death penalty, torture, and war. There is no provision for domestic violence or natural disasters, for example. In this regard, the ICRRA allows the Minister of Justice the leeway to grant SPS, or not, without the restrictions imposed by Art. 15. While the lack of clarity is problematic, it may also allow a more inclusive application of complementary protection.

One clear difference between the QD and the ICRRA is in Art. 15(a), which defines one of the grounds of serious harm as facing the death penalty. Tsourdi explains that 'according to international human rights law, countries that have abolished the death penalty may not return persons to countries where they will face the death penalty or execution' (Tsourdi 2014, p. 274). As Japan continues to employ the death penalty, it is unlikely there would ever be a case in which SPS was granted solely on the grounds that the applicant would face the death penalty if returned. Unsurprisingly, the ICRRA does not include any provisions related to this point.

Another difficult element in Art. 15 is the requirement that there be an 'individual threat to a civilian's life or person by reason of indiscriminate violence' (QD, Art. 15(c)). At first glance, requiring the threat to be 'individual' in the face of 'indiscriminate violence' appears contradictory in itself. More importantly, perhaps, is that if the threat is individual, it may be more appropriately determined that the applicant is a Convention refugee. Goodwin-Gill and McAdam described the requirement that the threat be individual as 'counter-intuitive to the very notion of violence that is random and haphazard' (Goodwin-Gill and McAdam 2007, pp. 327–28) and argue, instead, that Art. 15 was originally intended to protect 'persons fleeing indiscriminate effects of armed conflict or generalized violence without a specific link to Convention grounds' (Goodwin-Gill and McAdam 2007, pp. 326–27). It should also be said that the EU benefits from additional instruments that may be seen to complement the QD and provide protection where Art. 15 fails. The '2001 Temporary Protection Directive for those fleeing conflict en masse' (Goodwin-Gill and McAdam (2007, p. 328), for example, may provide a lesser level of protection without the caveat of the threat being 'individual'. Detailed discussion on the further stratification of

protection classes is beyond the scope of this paper, but it is enough to recognize that other instruments exist, and the QD is only a part of the wider CEAS. Japan is limited to one legal instrument for providing protections to meet its international obligations, but the lack of specificity in the ICRRA allows Japan to avoid the contradiction of Art. 15(c).

While the scope of this essay does not allow for an exhaustive discussion of Art. 15(c), it is important to note the growing body of jurisprudence in Europe that is gradually providing clarity on its interpretation. One important legal concept arising out of Europe's need to cope with Art. 15(c) is the 'sliding scale test'. Ippolito explains that 'the more the applicant could show that he was specifically affected by factors particular to his personal circumstances, the lower the level of indiscriminate violence required for him to be eligible for subsidiary protection', (Ippolito 2013, p. 261) and Tsourdi notes it emerged from the *Elgafaji* case (Tsourdi 2014, p. 277). Additionally, the *Sufi and Elmi* decision 'identified some specific (not exhaustive) criteria for assessing what the level of severity of a situation of general violence must be to reach the threshold of a 'real risk'' (Ippolito 2013, p. 266). Similarly, the *Tadic* case explained that 'the definition of non-international armed conflict encompasses situations where 'there is [...] protracted armed violence between governmental authorities and organised armed groups or between such groups within a State'' (Ippolito 2013, p. 276). All of these rulings help to clarify the meaning of the law in practice. There are, of course, other cases helping to establish standard practice in Europe, but it is important to note that even the UNHCR has stated 'the distinct ambit of Article 15(c) remains unclear' (Tsourdi 2014, p. 286). What is clear in Art. 15(c), and the relevant jurisprudence, is that applicants must demonstrate some degree of individual risk to be eligible for subsidiary protection.

There are far fewer cases available for public scrutiny in Japan, but some cases, from a brief summary provided by the Ministry of Justice, offer insight into Japan's view on the individual nature of threats from indiscriminate violence. Case 2 of the 2016 summary report explains that the applicant was not recognized as a refugee because persecution for a Convention reason was not established. However, SPS was granted due to conflict in the home country. The report notes that even though a ceasefire had been signed between the warring factions, 'the situation is unstable, as there are still reports of sporadic engagements and casualties' (Ministry of Justice Summaries 2016, Case 2). There is no evidence provided that the threat was individual but, rather, that the mere continuance of armed conflict, even sporadically, created enough risk to justify granting SPS. Similar grounds were mentioned in 5 among 12 case summaries provided in the 2015 report. The 2016 report offers 15 summaries, out of the 97 cases in which SPS was granted, so the available evidence provides a limited sample. However, six of these summaries show cases where SPS was granted specifically because of ongoing conflict in the country of origin, and none of them describe any need to establish a threat as being 'individual' in nature (Ministry of Justice Summaries 2016). The 2017 report shows three more cases in which SPS was granted due to conflict in the country of origin (Ministry of Justice Summaries 2017), as do the 2018 and 2019 reports (Ministry of Justice Summaries 2018 and 2019).

A particularly informative case is found in Case 2 of the 2018 summary report. In support of their claim for refugee status, the applicant noted that their family business had been attacked by government forces. However, the Ministry of Justice determined that 'the applicant's store was not individually targeted' (Ministry of Justice Summaries 2018, Case 2). This key finding led to the denial of refugee status with the Ministry of Justice clearly distinguishing between an individual threat amounting to persecution, but indiscriminate violence falling outside the scope of the 1951 Convention. The applicant was, however, granted SPS on the grounds that 'it was not possible to deny the possibility of being involved in a civil war' (Summaries 2018, Case 2). This offers a clear view into Japan's interpretation of indiscriminate violence versus individualized persecution.

Another example is found in Case 1 of the 2017 summary report. The applicant claimed refugee status on the grounds that he had been detained and beaten by a militia unit. The applicant was not recognized as a refugee because Japan's Ministry of Justice

determined that this was a kidnapping for ransom and did not fall under persecution based on the 1951 Convention. However, SPS was granted on the grounds that 'the battle between the home government and the rebels is still continuing' (Summaries 2017, Case 1). The original application was denied despite the individual nature of the threat because the threat was criminal and not persecutory. It is quite plausible for this case to be granted Convention refugee status in other jurisdictions. Be that as it may, SPS was granted due to indiscriminate violence with no mention of a component of individualized threat.

Case 1 from the 2018 summary report provides another example of SPS granted without consideration for whether the threat was individual in nature. The applicant requested asylum on the grounds of religious persecution. Japan failed to recognize the applicant as a refugee because there was no evidence of religious persecution by the State, including that the applicant had been issued a passport in his own name. However, SPS was granted on the grounds that armed conflict 'is still ongoing and unstable' and that 'it is acknowledged that there is no prospect of improving security' (Summaries 2018, Case 1). As in the cases from 2016 and 2017, the status was granted only for reasons of continued conflict, or indiscriminate violence, in the home country, and there are no indications that the threat was specific to the individual applicant.

Even recognizing the limited amount of information provided by the Japanese government on adjudicated cases, the information provided reveals no evidence that Japan requires the threat of indiscriminate violence to be specific to the individual. In this sense, the ICRRA, in practice, is a more inclusive model for granting complementary protection. Arguably, some of the cases granted SPS in Japan may have been recognized as a full-fledged Convention refugee in other jurisdictions. However, the existing case summaries reflect the possibility that Japan's SPS may provide international protection for a wider group of people than EU's subsidiary protection does.

### 4.1.3. The Role of Non-State Actors

It is worth mentioning that the QD specifies that non-State actors can cause, or pose the risk of causing, serious harm (QD, Art. 6(c)). This is significant as the 1951 Convention does not include any clear reference to non-State actors, and the issue has remained unsettled as to whether, in the case of refugee status, non-State actors could be the agents of persecution. The QD removes any doubt by specifying that persecution or serious harm from non-State actors, if the State is unable or unwilling to provide protection, makes one eligible for international protection.

The ICRRA makes no reference to non-State actors, nor does it explicitly require the agent of persecution or serious harm to be the State. It is left undefined. However, one can look to the existing jurisprudence to gain some insight into Japan's view. In *Afghan v. Japan*, the court recognized that 'in the civil war after 1992, the Hazaras had been the targets of military attacks by the Pashtuns and the Tajik groups', and that this meant 'it should be evident that the Hazaras were under the threat of persecution in Afghanistan' (Afghan v. Japan). This is a clear example of the Japanese court recognizing non-State actors as agents of persecution.

There are other cases, though, that reflect a different view. In Case 1 of the 2018 summary report discussed above, the Ministry of Justice claims to have denied refugee recognition because 'the applicant has never been harmed by the government of the country' (Summaries 2018, Case 1). This clearly indicates some degree of expectation that the government should be the agent of persecution. These significantly differing interpretations between the administration and the judiciary in Japan reflect the challenge of vague language. While vagueness allows flexibility, it also creates inconsistency. On the matter of non-State actors as agents of persecution, there is evidence that Japan has not fully developed a clear interpretation.

Article 6 of QD represents an important development in international protection by explicitly recognizing the potential for non-State actors to be the agents of persecution or

serious harm. As the ICRRA does not clearly address this issue, case law in Japan reflects differing and narrower views.

### 4.1.4. Push Factors vs. Pull Factors

One distinctive feature of SPS is that the Immigration Services Agency takes into account the conditions in Japan as well as conditions in the country of origin of the applicants. As discussed above, subsidiary protection in the QD is provided based upon the extent to which the individual, *if returned to the country of origin*, would face the real risk of serious harm arising from the death penalty, or torture, or armed conflict. In other words, it only takes into account the conditions in the country of origin, i.e., push factors. On the contrary, the case summaries made available by the Japanese Immigration Services Agency makes it clear that SPS has been occasionally granted based upon the extent to which the individual has been receiving medical treatment in Japan, or developed a family tie, or has been integrating in Japan, i.e., pull factors.

For instance, Case 14 of the 2016 summary report articulates that the case has been suffering from a disease for which treatment is unavailable in the country of origin, and that continuous treatment in Japan would be indispensable in light of official diagnoses submitted by a medical doctor. Such medical and health related reasons were also invoked for Cases 7, 10, 11, and 12 from the 2015 summary report.

Another reason often invoked is respect for family life. Case 15 from the 2016 summary report, Case 6 from the 2017 summary report, Cases 6 and 7 from the 2018 summary report, and Cases 4 and 5 from the 2019 summary report all provide the almost identical reason as follows:

> 'The statements provided by both spouses as regards their marital history and family life are generally consistent. The documents submitted by them demonstrate that the spouses live together and are supporting each other. In addition, the spouse has (a) Japanese child(ren) and their marriage appears to be stable and continuing. Given these reasons, it was thought to be necessary to allow his/her residence in Japan from humanitarian viewpoints.'

In terms of the level of integration in Japan, Case 5 from the 2018 summary report is noteworthy. The case is a minor living in Japan for 10 years, during which time she has attended Japanese schools, and thus it was acknowledged that her personality has been formulated in Japanese society and education. The summary report clearly acknowledged the level of her integration in Japan and concluded that the damage she would suffer in case of return to her country of origin would be serious.

It is true that such pull factors, including medical condition, family life and integration in the destination country have often been taken into account by European countries in light of Articles 2 (right to life), 3 (prohibition of torture), and 8 (right to respect for private and family life) of the European Convention on Human Rights (ECHR) (ECHR, 1950). in allowing residential status for asylum seekers. However, such consideration is outside the requirements provided for by the QD, and one distinctive element where Japan's SPS provides wider possibilities for protection than subsidiary protection under the QD.

### 4.2. Exclusions and Ending Protection

### 4.2.1. Exclusion Clauses

The QD requires exclusion from subsidiary protection in Art. 17. It is important that States are obligated to exclude persons falling under the categories of Art. 17, not merely allowed to exclude them. However, being excluded from subsidiary protection does not allow States to violate their international legal obligation of non-refoulement. McAdam has noted the important point that certain persons cannot be removed from a State due to the international obligation to avoid refoulement. She explains the difficulty of this in stating that it 'leaves standards of treatment for persons protected by non-refoulement but outside the scope of the [QD] to the whim of various States' (McAdam 2005b, p. 494). This is a further stratification of rights among protected persons that deserves further research

(See also Singer 2017). The point relevant to this section is that exclusion clauses in the QD require States to deny subsidiary protection on certain grounds but do not allow for the breach of non-refoulement, and this could potentially lead to the creation of additional groups of protected persons.

In the EU, persons are excluded from subsidiary protection eligibility on largely the same grounds as established in the 1951 Convention's Art. 1(F) (Refugee Convention, Art. 1F), but with a few differences. The acts common to both instruments are war crimes or crimes against humanity, serious crimes, or acts contrary to the purposes of the UN (QD, Art. 17(1)). The differences are nuanced but significant. Importantly, the 1951 Convention refers to 'non-political' serious crimes, but the QD omits this qualification, thus allowing a broader interpretation of the principle. This is significant in addressing subsidiary protection because the persons concerned are, by definition, not protected by the 1951 Convention. By omitting the caveat 'non-political', it is possible that those charged with political crimes could be excluded. Further additions to the 1951 Convention grounds for exclusion include that one may be ineligible for protection if they pose a danger to the community (QD, Art. 17(1)(d)), incite others to commit acts contained in Art. 17 (QD, Art. 17(2)), or those who are fleeing criminal prosecution for acts that would be considered criminal in the receiving State (QD, Art. 17(3)). The inclusion of security concerns allows a more restrictive application of subsidiary protection than the 1951 Convention allows for refugee determinations.

The ICRRA deals with exclusions in a starkly different manner. First, the ICRRA employs the 1951 Convention to determine exclusions from refugee status. Article 61-2(1) states that the 'Minister of Justice may, if an alien in Japan submits an application in accordance with the procedures provided for by a Ministry of Justice ordinance, recognize such person as a refugee (hereinafter referred to as "recognition of refugee status") based on the data submitted' (ICRRA, Art. 61-2(1)). There are no stipulations on exclusions beyond those included in the 1951 Convention. Second, unlike the QD, the ICRRA defines SPS in such broad terms that it does not appear mandatory to exclude anyone. As previously noted, there are no specific grounds enumerated on which SPS might be granted, so it is not surprising that that there are no clear grounds on which persons might be excluded. The case summaries provided by the Ministry of Justice do not include any cases in which SPS was denied. Therefore, it is not possible to investigate the manner in which exclusions may or may not be applied to the actual granting of SPS. This is an important topic in need of further research. What is important to note, in contrast with the QD, is that the ICRRA does not explicitly require exclusion from SPS on any particular grounds.

An important complexity arises from the ICRRA's application of exclusions to the granting of residential status rather than to refugee status determinations directly. Art. 61-2-2 describes the status of residence to be issued to recognized refugees as Long-term Resident and lists grounds on which one may be excluded from receiving that residential status. To clarify, these are exclusion clauses of the residential status for those already recognized as refugees and, therefore, are not applicable to the refugee recognition procedure itself. To rectify this apparent complication, one can refer to McAdam, who writes, '[a]lthough awarding a residence permit is a separate act from recognizing refugee status, in practice the two are generally linked, so that rights tend to flow from that point of residence as though it were the point of legal recognition of refugee status' (McAdam 2005b, p. 506). It is impossible to ignore this link, especially when the ICRRA mentions no other forms of exclusion related to SPS. The QD specifically defines exclusion clauses in Art. 17 as pertaining to those seeking subsidiary protection while the ICRRA only lists possible exclusions from the granting of a particular residential status for Convention refugees. Due to the ICRRA's lack of clarity regarding SPS, further research is needed to ascertain whether or to what extent these exclusions apply to SPS, or whether the exclusions affect the type of residential status to be granted.

4.2.2. Internal Protection Alternatives (For Comprehensive Research on IPA, see Schultz 2019)

An additional exclusion may be found in Art. 7 and 8 of the QD which, for this discussion, should be read together. Art. 8 excludes from eligibility those who have access to internal protection in any part of the country of origin (QD, Art. 8), and references Art. 7 as providing the criteria on which to determine if such protection is available. Specifically, Art. 7 defines the actors considered capable of providing protection and the manner in which that protection must be evaluated. As the notion of internal protection alternatives (IPA) (Schultz 2019) potentially makes certain applicants ineligible for subsidiary protection, it is worth discussing.

Gil-Bazo notes that Art. 7(2) of the QD (QD, Art. 7(2)) is controversial in 'the understanding that refugee status may not arise when an internal flight alternative exists (Art 8) and when protection can be provided by non-state actors (Art 7(1))' (Gil-Bazo 2006, p. 1). More specifically, this allows exclusion from refugee status or subsidiary protection if an internal protection alternative exists, even from a non-state actor, so long as that entity takes 'reasonable steps' to provide protection, but without regard for whether that protection is effective (Gil-Bazo 2006, p. 1). The ICRRA does not mention any allowances for IPA, or for the potential of non-state actors to provide protection. While this lack of clarity may lead to inconsistent applications and is generally considered undesirable, the absence of this provision in the ICRRA means applicants and adjudicators need not debate internal alternatives but, rather, can limit the discussion to whether the State is able or willing to provide effective protection. It should also be noted that the QD has created unnecessary vagueness by including both an ambiguous term like 'reasonable steps' in Art. 7(2), and the possibility that protection might come from non-state actors in Art. 7(1) (QD, Art. 7(1)). This issue has been widely explored by Eaton who, like Gil-Bazo, criticizes the idea, and notes Hathaway and Foster's objections. To succinctly summarize the issue, Eaton quotes de Moffarts as explaining that '[t]he reasonableness approach tends to an eclectic or ad hoc jurisprudence concerning claimants from the same countries and in similar situations' (Eaton 2012, P. 774). Status determination procedures, like any application of law in a just society, must be administered equally and uniformly. Subjective language in legal instruments that creates 'eclectic or ad hoc jurisprudence' is unacceptable. If Japan's ICRRA is to be criticized for failing to offer clarity in many regards, the QD must be seen as flawed in the wording of Art. 7, as well.

The next logical issue to address is how to determine whether the protection is effective. Art. 7(2) of the QD argues that an appropriate entity need only 'take reasonable steps to prevent the persecution or suffering of serious harm' in order to be effective. As this research is focused on complementary/subsidiary protection, the key factor will be protection from serious harm. Hathaway and Foster argue the need for clearer criteria for judging whether IPAs are effective, but ultimately assert that the absence of effective State protection should be enough to trigger international protection, either through the 1951 Convention, or some form of subsidiary protection (Eaton 2012, p. 772). Basic criteria including legality, protection against refoulement, and affirmative State protection must exist for protection to be deemed effective. In all but the most exceptional cases, the State is the only entity capable of meeting these criteria. Further, tests based on 'real chance' or 'serious possibility' of suffering serious harm are more appropriate than whether the State has taken reasonable steps to provide protection. If the State cannot provide a level of protection where there is less than a 'real chance' or 'serious possibility' of serious harm, then there is no effective protection. For the purposes of determining subsidiary protection, the absence of effective protection should be grounds for granting it.

By investigating the available information on SPS cases in Japan, one can surmise whether the Ministry of Justice is considering IPAs and, also, whether it requires that a State provide effective protection or merely that it takes reasonable steps to provide protection. Case 2 of the 2016 report offers some insight. Though caught up in internal conflict, the applicant was not recognized as a refugee because it was determined that the

indiscriminate violence of the conflict did not qualify as persecution. The applicant was, instead, granted SPS. There is evidence that IPAs were not a factor in the decision because the report acknowledges that while there were areas of the home country that were not engaged in the conflict, it was not reasonable to expect the applicant to seek protection there. There is also evidence that Japan considered the effectiveness of the protection rather than only taking reasonable steps to provide protection. The report notes the State and the warring factions had signed a ceasefire, but the Immigration Bureau granted SPS to the applicant because the situation remained unstable. Having signed a ceasefire may be considered a reasonable step toward providing protection under the EU's QD, but the instability of the situation implies the protection was not effective. According to the Ministry of Justice report, 'the possibility could not be denied' that the applicant would risk 'life and serious hazard' if returned (Summaries 2016, Case 2).

Case 3 from the 2017 report offers another example. The applicant claimed refugee status on the grounds he was a persecuted minority. Japan failed to recognize his refugee status because information on the home country, it was deemed, showed that such discrimination was not common or accepted by the State. SPS was granted on the grounds of internal conflict, but the Ministry of Justice included insights into its view on IPAs. The report states '[i]t is recognized that if the case returns home, s/he will live in State B, and s/he may be involved in the battle between government forces and rebels in the state' (Summaries 2017, Case 3). This ruling implies there is no expectation that the applicant should seek an IPA, even though it recognizes that other parts of the country of origin may be safe.

Another example from the 2017 report, Case 4, offers even further insight. A woman applied for refugee status because she had refused to marry the chief of her village and feared being killed on return. The asylum application was denied on the grounds that the State did not condone such actions and that the Constitution of the home country protected women's rights. However, SPS was granted on the grounds that the woman was likely to face long-term abuse from the chief if she returned (Summaries 2017, Case 4). This ruling indicates that the State is expected to be the agent of protection, and may be seen to imply that only the State can be the agent of persecution while also allowing that non-State actors may be the agents of serious harm. As the agent of serious harm in this case was a local chief, the Ministry of Justice might have considered that the applicant could seek an IPA in another part of the country. There is no indication this effected the decision.

Based on the available information, it appears Japan's interpretation focuses more on the role of the State in protection or persecution within the country of origin. There is clearly a shortcoming of the QD in the language found in Art. 7(2). The more inclusive and consistent approach is to expect the State to be the source of protection and requiring that protection be effective.

### 4.2.3. Cessation

Exclusion clauses may be said to be punitive in nature, denying protection based on the actions of the applicant. Cessation, on the other hand, should be seen as reflecting a change in the circumstances that created the protection need. The QD sets out cessation criteria specifically for recipients of subsidiary protection, separate from the cessation elements applied to refugees. Art. 16 establishes that subsidiary protection will end if and when the circumstances that made the applicant eligible have changed (QD, Art. 16). The article also includes caveats about the significance and permanence of the change, as well as the possibility that previous serious harm may make a person unwilling or unable to avail themselves of protection in the country of origin. The ICRRA does not clearly distinguish between cessation and revocation of refugee status, except to say that Art. 2-3 (ii) invokes the entirety of the 1951 Convention in defining who is a refugee. However, it does not contain any provisions for the cessation of SPS. Art. 61-2-7 includes clauses for both cessation on the grounds that circumstances have changed, and revocation for punitive reasons, only concerning the Convention refugee status (ICRRA, Art. 61-2-7-1(ii),

61-2-7-1(iii)). The latter will be discussed in the following section. Art. 61-2-7(ii) simply states that refugee recognition may be revoked if the 'alien has come to fall under any of the cases listed in Article 1, C-(1) to (6) of the Refugee Convention' (ICRRA, Art. 61-2-7(ii)). Those include voluntary repatriation, the acquisition of a new nationality, or a change in the circumstances on which he/she was recognized as a refugee (Refugee Convention, Art. 1C).

The key difference in the QD and the ICRRA is in their respective relationship to the 1951 Convention. The QD is somewhat more generous than the 1951 Convention in stating that cessation 'shall not apply to a beneficiary of subsidiary protection status who is able to invoke compelling reasons arising out of previous serious harm for refusing to avail himself or herself of the protection of the country of nationality or, being a stateless person, of the country of former habitual residence' (QD, Art. 16(3)). The ICRRA, as noted above, simply references the 1951 Convention grounds for cessation and exclusion, without clarifying cessation or exclusion grounds for SPS. Hathaway and Foster, in quoting Grahl-Madsen, argue that adding 'cessation grounds into the exhaustive list contained in Art 1C of the Convention is a legally impermissible course' (Hathaway and Foster 2014, p. 463). Both approaches respect this view of international legal standards, although the QD appears to offer broader protection than the ICRRA.

### 4.2.4. Revocation

The QD addresses revocation of subsidiary protection in Art. 19. The grounds listed are cessation under Art. 16 (QD, Art. 19(1)), if it comes to be known the applicant should have been excluded under Art. 17 (QD, Art. 19(2)), subsidiary protection was granted based on deceit (QD, Art. 19(3)(b)), or the applicant fails to provide information as required in Art. 4 (QD, Art. 19(4)). Interestingly, the requirement to revoke subsidiary protection status based on the applicant being deceitful is qualified. Art. 19(3)(b) includes the caveat that the misrepresentation must have been 'decisive for the granting of subsidiary protection status' (QD, Art. 19(3)(b)). This leaves open the possibility for violations to be forgiven under certain circumstances, even though Art. 19 imposes an obligation to revoke status if any of the criteria apply.

ICRRA Art. 61-2-7 defines the grounds on which refugee status may be revoked (ICRRA, Art. 61-2-7), and Art. 61-2-8 establishes how the residence status issued based on refugee status recognition will subsequently be revoked. Neither of these explicitly applies to SPS, but the ICRRA fails to mention any means by which SPS may be revoked. The grounds on which refugee recognition can be revoked, in addition to the cessation clause already discussed, include if the applicant was recognized 'by deceit or other wrongful means' (ICRRA, Art. 61-2-7-1(i)), or if the person commits acts described in Art. 1F(a) or 1F(c) of the 1951 Convention (ICRRA, Art. 61-2-7-1(iii)). Art. 61-2-7 (iii) even states that refugee recognition may be revoked if the refugee *upon recognition* committed crimes specified in Art. 1 F (a) or (c) of the Refugee Convention (ICRRA, Art. 61-2-7-1(iii)). These are normally considered exclusion clauses in refugee status determination, and the ICRRA indeed does employ the same exclusion grounds (as well as cessation grounds) by its Art. 2-3 (ii) adopting the entire Art. 1 of the Refugee Convention when defining the term refugees for ICRRA. However, re-invoking the same exclusion grounds for revocation after the fact might well be noteworthy. Returning to the value of the overarching terminology in the QD of 'international protection' as a category encompassing both refugee status and subsidiary protection, one can see that the ICRRA fails to provide a mechanism specifically for the revocation of SPS. It is possible that residence status could be revoked through Art. 22-4 (ICRRA, Art. 22-4), which covers revocation of residence status generally, but the ICRRA fails to articulate whether revoking residence status negates or otherwise impacts SPS. Importantly, refugees are explicitly protected from having their residence status revoked under Art. 22-4 (ICRRA, Art. 22-4(1)), with Art. 61-2-7 and Art. 61-2-8 creating separate criteria for those recognized as refugees. The ICRRA provides no clarity on whether recipients of SPS should be treated similarly to refugees. However, given the fact that SPS

is created within the article addressing refugee recognition, and that it only applies to those whose refugee status has not been recognized, it may be that some link exists. This is an area in need of further clarification as well as research.

*4.3. Content of Protection*

This section will explore the content of the protection offered under the QD and what is offered through SPS. Chapter VII of the QD codifies a list of the minimum rights owed to anyone receiving international protection. In emphasizing that these are minimum standards, Storey writes that 'on [Chapter VII] the harmonization can only ever be upwards, not downwards' (Storey 2008, p. 19). Art. 20(2) states explicitly, 'This Chapter shall apply both to refugees and persons eligible for subsidiary protection unless otherwise indicated' (QD, Art. 20(2)). This treatment of all persons receiving protection as one category reflects an effort to limit the differentiation of rights between separate protection classes and, as noted previously, is noticeably absent from the ICRRA.

Guild rightly points out that the CEAS is committed to respecting both human dignity and the right to asylum (Guild 2015), and that the minimum standards in Chapter VII represent 'an explicit acknowledgement that material conditions of living above a commonly agreed threshold are a central component of the right to dignity' (Guild 2015, pp. 500–1). These standards include protection against refoulement (QD, Art. 21), family reunification (QD, Art. 23), access to employment (QD, Art. 26), access to education (QD, Art. 27), opportunities to validate foreign qualifications (QD, Art. 28), access to social welfare assistance (QD, Art. 29), access to healthcare (QD, Art. 30), access to housing (QD, Art. 32), freedom of movement (QD, Art. 33), access to integration services (QD, Art. 34), and assistance in voluntary repatriation (QD, Art. 35). There are other elements to Chapter VII that will be discussed in a later section on procedural issues, but this list generally enumerates the minimum reception conditions owed to anyone receiving international protection in the EU. The common caveat running through Chapter VII is that these standards must, broadly speaking, be provided at the same level as it is provided to nationals. Guild points out that the QD marks the first time Europe has required material support such as housing, and that Member States are not explicitly required to provide any of these standards to EU citizens, only for third-country nationals under international protection (Guild 2015, p. 500).

As noted previously, Japan's ICRRA does not include an overarching category of protected persons in the way the QD employs the term international protection. Therefore, one must look separately at the reception conditions for recognized refugees as compared to those granted SPS. Hashimoto notes refugees 'are, unsurprisingly, guaranteed the widest rights and entitlements among all forced migrants in Japan' (Hashimoto 2019, p. 131). It is important to note that, even for those recognized as refugees most of the entitlements they enjoy are offered under other relevant domestic legislations regarding welfare, employment, healthcare and education, but they are not codified as rights in the ICRRA. Reflecting Art. 23 of the QD, refugees are entitled to family reunification, though 'family members remaining in the country of origin are required to go through a visa application process' (Hashimoto 2019, p. 132). Refugees also have access to healthcare, pension, social welfare programs, public housing, employment, and education at the same level as Japanese nationals (Hashimoto 2019, p. 132). Hashimoto does note one significant gap in Art. 26 of the Japanese Constitution related to education when she writes 'Japanese parents (or guardians) are legally obligated . . . to ensure that their Japanese children attend school through the junior high school level', but that this clause 'has been interpreted so that foreign parents do not have the same legal obligation' (Hashimoto 2019, p. 132). For the sake of the child this issue needs to be addressed. However, for the purposes of this commentary, it is enough to say that refugees have access to education.

Another valuable service offered to refugees is the government-funded integration assistance course from the Foundation for the Welfare and Education of the Asian People (FWEAP). This program provides '572 hours of Japanese language training, 120 hours of daily life orientation, assistance in job-hunting, and employment training opportunities'

(Hashimoto 2019, p. 133). Additionally, through this free course 'refugees are provided free accommodation, free nursery care, daily subsistence allowance, transportation costs, medical costs during the course, and a one-time relocation assistance grant' (Hashimoto 2019, p. 133). This program is clearly of great value to refugees, aiding them immensely in the integration process and providing much-needed material support during the process. However, this program is only available to recognized or resettled refugees, not to any other category of immigrant in Japan. This means that those granted SPS are excluded from these valuable services, even though their material needs may be largely the same as those with recognized refugee status.

In contrast to the comprehensive services offered to refugees, those protected under SPS receive a noticeably lower standard of reception. Family members may apply for a visa, 'but there is no preferential consideration or special assistance for them' (Hashimoto 2019, p. 136). SPS recipients do have access to the national healthcare and pension schemes, but their access to social welfare assistance is questionable. As Hashimoto explained, 'decisions [on public cash assistance] often vary depending on municipality (or individual officials)' (Hashimoto 2019, p. 136). Like the vagueness in all aspects of the ICRRA, the failure to codify reception standards has led to ad hoc decisions that are inherently inconsistent. In principle, those granted SPS have access to education, housing, and employment at the same level as other legally staying foreigners in Japan, not at the level of nationals. In terms of residential status, those granted SPS are usually given 'designated activities' (*tokutei katsudō*) status for (renewable) one year, with a legal pathway to naturalise eventually by applying for Japanese nationality, although no preferential considerations given to Convention refugees are available for SPS recipients. The question of naturalisation presents an inter-generational conundrum to SPS holders and their children born in Japan, given the persistent denial of dual nationality, the societal discrimination against foreigners in general, and the lack of anti-discrimination laws in Japan. Although the authors are not aware of hate crimes specifically targeted at the SPS holders, it is not uncommon, for instance, for landlords to refuse to rent to foreigners, regardless of residential status or financial means (author's personal experience). While more detailed analysis on the content of protection available for SPS holders is conducted elsewhere (Hashimoto 2019), suffice it to say that their access to social welfare and other rights and entitlements could be unnecessarily precarious, because the ICRRA fails to establish explicitly what type of residential status should be given to those under SPS.

Again, the ICRRA's greatest weakness, lack of clarity and specificity, is sharply contrasted with the details provided in the QD. The EU mandates that Member States provide a minimum standard of reception for anyone receiving international protection, including subsidiary protection. In contrast, the ICRRA, which only regulates border control, does not codify reception requirements. Instead, other domestic instruments in relation to pension, child welfare, social insurance, and labour ensure Japan is providing needed services to refugees, although some of the services are not necessarily guaranteed to those granted SPS.

*4.4. Procedural Issues*

4.4.1. Application Procedures

One procedural difficulty many claimants will face in the application process is the language barrier. It is worth noting that Art. 22 of the QD requires States to provide anyone granted international protection 'with access to information, in a language that they understand or are reasonably supposed to understand, on the rights and obligations relating to that status' (QD, Art. 22). There is no equivalent requirement in the ICRRA, and lawyers representing the claimants are usually responsible for providing translations of all documentation, which is required to be submitted in Japanese. This must be seen as a shortcoming.

Art. 4 of the QD is titled 'Assessment of Applications for International Protection' (QD, Art. 4). As noted above, this language applies to both refugee status and subsidiary

protection, reflecting the fact that applicants need not file separate requests. Subsidiary protection will be offered when the applicant fails to meet the requirements of refugee status but does face the risk of serious harm if returned to the country of origin. Art. 4 offers a detailed explanation of how refugee status will be adjudicated, specifically enumerating the responsibilities of the applicant to provide information, and the manner in which the receiving country should evaluate the case. Like the QD, the ICRRA allows for the granting of SPS after an application for refugee status has been denied, and there is no need for an applicant to request it. SPS is only granted after an application for refugee status has been denied, so the elements related to application procedures are equally relevant to refugee status and SPS, just as they are under the QD. There are several elements of the administrative process that deserve discussion.

Both the QD and ICRRA include reference to the timeliness of applications, but in significantly different ways. The ICRRA stipulates that the applicant must apply for refugee recognition within six months of their arrival, or from the time they became a refugee (in the case of sur place), unless there are 'unavoidable circumstances', if one is to be automatically granted the long-term residential status upon refugee recognition (ICRRA, Art. 61-2-2(1)(i)). The QD simply requires the applicant provide 'as soon as possible all the elements required to substantiate the application for international protection' (QD, Art. 4(1)). Japan's separate treatment of refugee recognition and the granting of residential status, again, adds a layer of complexity to the issue. In the previous immigration law, a more restrictive 60-day rule was applied, as indicated above. While that requirement was revoked as part of the 2004 revisions, there remains a six-month time limit that effects provisional stay permits (kari taizai) and long-term residential status, although it does not influence the refugee status determination or SPS directly. Due to the absence of any case summaries in which SPS has been denied, it is not possible to ascertain whether the timeliness of an application has had any bearing on the decision. The wording of the QD may reflect a recognition of the obstacles applicants may face, while that of ICRRA generally appears to put more emphasis on timeliness in asylum applications.

In addition, and of particular interest to this research, Art. 4(5) of the QD enumerates several grounds on which the evaluator of an application should excuse the absence of documentary evidence. Taking into account the difficulty refugees and asylum seekers may face in producing certain documents or other tangible evidence, this section of Art. 4 requires adjudicators to consider whether the applicant has made a genuine effort, can explain why certain elements are missing, offers 'coherent and plausible' statements, has applied in a timely fashion or given the reason why they did not, and whether their statements are credible (QD, Art. 4(5)). This is an important inclusion in the QD as it respects the difficulties of producing tangible evidence when forced to migrate. In Japan, the ICRRA lacks such a detailed explanation of the assessment process. Art. 61-2 only requires that an applicant 'submits an application in accordance with the procedures provided for by a Ministry of Justice ordinance' (ICRRA, Art. 61-2(1)). It is true that the Ministry of Justice provides greater detail on the application form, including instructions on information and documentation required (Immigration Services Application), but there is no requirement the refugee inquirers make allowance for missing evidence in specific circumstances. Additionally, the lack of a legislated mechanism to define the application process leaves open the possibility for changing policies, or forms, almost arbitrarily. The roles and responsibilities of the refugee inquirers are explained in Art. 61-2-14 of the ICRRA, but it is noted only broadly that they 'may inquire into the facts, if necessary for the recognition of refugee status', and that they may request documentation or ask questions of the applicant (ICRRA, Art. 61-2-14). As with other elements of the QD, the key difference of the ICRRA is its failure to enumerate specific rights and responsibilities. Japan has opted for a less codified and, arguably, more flexible system that gives the Ministry of Justice a great deal of discretion in how cases are evaluated. While this element of the ICRRA lacks any acknowledgement of the challenges faced by applicants, it should be noted that *Turkish v. Japan* provides some evidence that Japanese courts might be sympathetic to this issue.

In asserting that the burden of proof was on the applicant, the Nagoya District Court also acknowledged the difficulties applicants may face in obtaining evidence, declaring 'the ultimate decision should be made on the basis of the credibility of his/her core arguments, also in the light of their consistency and reasonableness' (*Turkish v. Japan* 2004, p. 24). Although this recognition is not codified in the ICRRA, it reflects a view largely in line with that found in the QD.

### 4.4.2. Assessments in Practice

This section focuses on assessment in terms of refugee status determinations. As SPS is a discretionary mechanism, the ICRRA does not specify a process of assessment. The assessment threshold for SPS, then, must be inferred from the refugee status determination procedure. In this regard, it is important to note the challenges applicants have faced in the assessment process in Japan. The ICRRA does not define extensive or exhaustive parameters for the process, so one must look to the procedure in practice to see some of the shortcomings. Arakaki has correctly argued that in refugee status determination procedures 'the human rights at stake are fundamental ones' and that, therefore, the determination procedures 'should be based on the highest standards of fairness' (Arakaki 2016, p. 80). He then goes on to enumerate several serious concerns with the Japanese system. 'The issues include the opportunity to be heard, adequate time for preparation, access to counsel, the opportunity to confront adverse evidence, reasons for decision, the opportunity for repeal or review, and the benefit of the doubt' (Arakaki 2016, p. 80). Other problems have been documented by Yukari Ando, particularly in relation to the Japanese courts interpretations of persecution that seem to be well outside of international practices. Ando quotes one court as ruling that 'if forced labour occurs just once a week, it is acceptable' (Ando 2016, p. 41). Any objective observer must agree that forced labour constitutes a gross violation of fundamental human rights and surely amounts to persecution. The issues identified by Arakaki and Ando may well contribute to the incredibly low number of refugees recognized in Japan's determination procedures. Again, these issues have been raised in reference to refugee status determination, and it is only by inference that one can surmise that these issues may also pertain to assessments of SPS.

### 4.4.3. Appeals

Importantly, the QD does not define the parameters for appealing a decision on status determination, but there are mechanisms provided in other instruments of the CEAS, the domestic legislation of Member States, and through the European Court of Human Rights. Japan, on the other hand, directly establishes the right to appeal refugee status determinations in Art. 61-2-9 of the ICRRA. Through this mechanism, applicants can file objections to denial or revocation of refugee status. One point of criticism in this system is that the Minister of Justice is responsible for administrative appeals. Amnesty International has complained that the same office who made the initial decision should not also be responsible for hearing the appeal (Amnesty International 1993). However, Yamagami responded that 'in Japan any administrative decision can be brought before the courts, which are totally independent from the administration' and that 'decisions of a court can be appealed to a higher court' (Yamagami 1995, p. 64). In the Japanese system of refugee status determination procedures, the first administrative appeal is normally heard by Refugee Examination Counselors (REC) since 2005 (ICRRA, Art. 62-2-10), but applicants also have the option of appealing to the courts. RECs 'are persons of reputable character who are capable of making a fair judgment on the appeal, and who have an academic background in law or international affairs' (Ando 2016, p. 41), but there is no requirement that they be experts in refugee law. While this seems to be a reasonable defense of the practice, Arakaki has argued that the most important factor in the appeals process is whether it 'reasonably and effectively achieves the purpose of the [1951] Convention' (Arakaki 2016, p. 93). In relation specifically to SPS, it is true that nothing prevents the RECs from recommending that the asylum applicant should be granted SPS. One significant limitation of the REC

system, however, is that their views are simply recommendations for the Minister of Justice to consider, without any legally binding force. In other words, the Immigration Service Agency could overturn the RECs' view be it refugee recognition or granting of SPS and still deny any protection for the applicant. It is also true that the right exists for claimants to appeal administrative decisions in the courts, and such appealing to the courts offers a clear, independent path of appeal for those denied refugee status or SPS. However, because SPS is only granted at the discretion of the Minister of Justice, there is no clear argument to be made that one has been unjustly denied. Here again, the lack of clarity surrounding eligibility criteria for SPS hampers the effectiveness of legal recourse to the courts in Japan.

Another problematic concern raised by the ICRRA's lack of clarity is that, in a practical sense, some legal counsels representing asylum seekers suspect that the Minister of Justice issues SPS when asylum applicants denied the Convention refugee status are about to appeal to the court. If accurate, this would mean the Minister uses SPS as a tool to deter applicants from pursuing judicial appeal procedures. Such a deterrence function potentially played by SPS is another area that requires further research from the viewpoints both of the Immigration Services Agency and of asylum applicants.

### 4.4.4. Issuance of Resident Permits

The EU and Japan address the issuing of resident status, or permits, in similar ways. The QD addresses this requirement in Art. 24. Here, one can see that Member States are required to issue refugees with permits no less than three years (QD, Art. 24(1)), but those under subsidiary protection are only required to be issued permits of at least one year (QD, Art. 24(2)). The ICRRA mandates that recognized refugees (as well as resettled refugees) should be given the status of Long-term Resident (ICRRA, Art. 61-2-2), which is valid for (renewable) five years (Hashimoto 2019, p. 131). There is no stipulation as to the type of status that must be granted to those under SPS. As with many elements of SPS, it is left to the discretion of the Ministry of Justice. As previously noted, SPS recipients are normally issued the status of 'designated activities', which is usually valid for (renewable) one year, although there is no guarantee that it would be renewed and the SPS status could be lawfully revoked for various reasons (Hashimoto 2019, p. 135). The failure to codify this, again, leaves room for confusion and ad hoc decisions, although, in practice, Japan treats residence permits in a similar fashion to the EU. It is detrimental particularly for those granted SPS that these practices are not enshrined in the law.

### 4.4.5. Travel Documents

In adherence to the requirements of the 1951 Convention (Refugee Convention 1951, Art. 28), both the QD and the ICRRA require the issuance of travel documents to recognized refugees, but only the QD extends this to those granted subsidiary protection. Art. 25(2) of the QD specifically states that those under subsidiary protection must be issued travel documents if they lack other appropriate documents that would allow international travel (QD, Art. 25(2)). Art. 61-2-12 of the ICRRA covers the issuance of travel documents and only requires they be issued to recognized refugees (ICRRA, Art. 61-2-12). While there is seemingly no prohibition on issuing travel documents to those granted SPS, there is no law requiring it. Whenever those with SPS need to travel abroad from Japan, the Immigration Services Agency issues a re-entry permit upon request, which may facilitate consideration of entry by the authorities of potential destination countries, albeit with no guarantee (Hashimoto 2019, p. 135). As with other aspects of the ICRRA, the failure to extend this access to official travel documents to SPS recipients further disadvantages those protected persons not recognized as refugees.

### 5. Conclusions

This article examined Japan's SPS under the ICRRA in light of subsidiary protection codified under the QD and elucidated several strengths and weaknesses of Japan's complementary protection for those in need of international protection. One common theme

throughout this exercise is the need for clarity and specificity to ensure fair and consistent application of the law. This means specificity in the law, not in a restrictive definition of persons in need of protection.

Careful examinations conducted in this research elucidated several strengths of the Japanese system. Such strengths typically emanate from the vagueness of standards that could lead to flexibility and inclusiveness in certain areas. In terms of eligibility criteria, the vagueness of the ICRRA allows for a more inclusive application of protection status, while the QD offers a restrictive set of criteria. A few case summaries demonstrated such inclusiveness in questions regarding IPA and non-State actors, although the jurisprudence in Japan indicates that the matter is not settled for SPS. If decisions were always made magnanimously with the best interests of the applicant as the singular goal, the Japanese SPS system might be preferable to the European system. This research also demonstrated that Japan is more willing to consider the individual circumstances of claimants in relation to their genuine connections in Japan. As the cases discussed illustrate, family relationships, social integration, and the well-being of children are considered by the Ministry of Justice as positive factors when deciding whether to grant SPS. It appears that family structure, with particular concern for stability and education for children, influences these decisions. In this way, one can see that while the QD places exclusive emphasis on the push factors (risk of serious harm in the country of origin), Japan pragmatically accepts the reality of pull factors (establishment of roots within the host country).

Meanwhile, the SPS under ICRRA demonstrates a number of fundamental weaknesses. One of the most notable is the complete absence of the eligibility requirements, as the retired immigration officials themselves admitted and regretted in their interviews. The ICRRA leaves the decision entirely to the discretion of the Minister of Justice whether or not to grant the status. Lack of specificity risks an inconsistent application of the law, resulting in ad hoc decisions and depriving the applicants of any grounds for appeals. The same analysis applies to exclusion, revocation, and procedural clauses. Due to the historical background of SPS as discussed above, the ICRRA leaves SPS in a precariously unclear area between Convention refugees and other categories of immigrants. If SPS is to be considered a genuine form of subsidiary protection, there needs to be more clarity in all areas. Europe, by contrast, benefits from clarity and specificity by enumerating the grounds on which subsidiary protection must be granted, revoked, or denied, thus ensuring a more consistent application in practice.

In a similar vein, the QD codifies the minimum standards for reception conditions for both refugees and those under subsidiary protection. The detailed enumeration of these standards ensures a basic respect for human dignity, and it is significant that these are enshrined in a binding legal instrument. Japan, by contrast, has included very little in the ICRRA to guarantee the reception conditions of refugees, and nothing at all with respect to those under SPS. It would be important that Japan specifically enumerate the assistance to be guaranteed to those granted SPS.

In terms of international relevance and wider implications of this study, the ongoing efforts to develop and improve the complementary protection scheme in Japan reflect the global trend of complementary protection emerging as an increasingly important component of international protection. The evolving system in Japan, as highlighted in this research, will surely contribute to the growing international body of law related to international protections for those falling outside the 1951 Convention definition of refugee. With the substantial body of scholarship already devoted to complementary/subsidiary protection broadly, and to the QD specifically, it is hoped that this work on Japan's approach to the issue will add to the understanding of how such protections are essential in light of the restrictions of the 1951 Convention definition and will offer new insights into how these protections can be offered. On a practical front, it is hoped that such studies will gradually lead to international standardisation of subsidiary/complementary protection schemes, so that equal treatment could be made available for forced migrants anywhere in the world, which would facilitate fair burden-sharing among countries of asylum. On an

academic front, such studies might also help to promote further research and debate into this emerging field of international law.

In this regard, this study has illuminated several areas for further research into refugee law in Japan. Several specific questions have been raised. First, more research is needed into the relationship between exclusion clauses and the granting of SPS. The ICRRA only explicitly links these to refugee status, and the effect they have on SPS is unclear. Along these lines, further research is needed into cases where refugee status is recognized and/or SPS is granted but exclusions are invoked to affect the status of residence accorded. Additionally, further research is needed to clarify whether and to what extent Art. 22-4 can be used to revoke the residential status of someone granted SPS. There is also a need for more work on the appeals process in Japan, specifically on what grounds an applicant might appeal SPS decisions in light of the absence of specificity in the law. Further, related to appeals, more insight is needed into whether and to what extent the Ministry of Justice grants SPS as a means to deter appeals of refugee determination procedures. Last but not least, it is hoped that the ongoing discussions on the new amendment bill to ICRRA would lead to a formal and full-fledged establishment of subsidiary protection in Japan and more data on jurisprudence would become publicly available, so that truly comprehensive analysis could be conducted based on exhaustive information in the future.

In conclusion, this research demonstrates that Japan has offered international protection to those who fall outside the parameters of the 1951 Refugee Convention, to a certain extent, by granting SPS. The court rulings and case summaries from the Ministry of Justice reveal some similarities between the Japanese protection scheme and its counterpart in Europe and even a few advantages as compared to the European subsidiary protection scheme. Fundamentally, the ICRRA should provide more clarity, specificity, and consistency for those in need of complementary protection. Such expanded protection could be extended by Japan through creating a more robust regime for subsidiary protection by clarifying and solidifying the legal framework and practice surrounding SPS that has existed for nearly 30 years. In the process, Japan has the opportunity to use the QD as an example and to correct the shortcomings evident in Europe's scheme.

**Author Contributions:** All authors have read and agreed to the published version of the manuscript.

**Funding:** The research received no external funding.

**Institutional Review Board Statement:** The study was conducted according to the guidelines of the Declaration of Helsinki, and approved by the Ethics Committee of the Refugee Law Initiative at the University of London. (approval number SASREC_1819-359-MA dated 1 May 2019).

**Informed Consent Statement:** Informed consent was obtained from all subjects involved in the study.

**Data Availability Statement:** Not applicable.

**Conflicts of Interest:** The authors declare no conflict of interest.

## Appendix A

### Legal Instruments

Act No. 62 of 1958, Migration Act 1958—Volume 1 [Australia]. 8 October 1958. Available online: https://www.refworld.org/docid/4e23f3962.html (accessed on 11 October 2019).

DIRECTIVE 2011/95/EU OF THE EUROPEAN PARLIAMENT AND OF THE COUNCIL of 13 December 2011 on standards for the qualification of third-country nationals or stateless persons as beneficiaries of international protection, for a uniform status for refugees or for persons eligible for subsidiary protection, and for the content of the protection granted (recast) [2011] OJ 2 337/9.

Council of Europe, European Convention for the Protection of Human Rights and Fundamental Freedoms, 4 November 1950, ETS 5, 3 September 1953. Available online: https://echr.coe.int/Documents/Convention_ENG.pdf (accessed on 19 September 2019).

Immigration Control and Refugee Recognition Act (Cabinet Order No. 319 of 1951). Available online: http://www.cas.go.jp/jp/seisaku/hourei/data/icrra.pdf (accessed on 19 September 2019).

Immigration Control and Refugee Recognition Act (Japan) Revisions 15 July 2009. Available online: http://www.immi-moj.go.jp/english/newimmiact/pdf/RefugeeRecognitionAct01.pdf (accessed on 7 September 2019).

Immigration Services Agency of Japan, Application for Refugee Status. Available online: http://www.immi-moj.go.jp/english/tetuduki/kanri/shyorui/07.html (accessed on 20 September 2019).

UN General Assembly, Convention Relating to the Status of Refugees (28 July 1951) UNTS 189, 137. Available online: https://www.refworld.org/docid/3be01b964.html (accessed on 7 September 2019).

United Nations, Statute of the International Court of Justice (18 April 1946) Art. 38. Available online: https://www.icj-cij.org/en/statute (accessed on 15 December 2020).

United States: Immigration and Nationality Act (last amended March 2004) [United States of America], 27 June 1952. Available online: https://www.refworld.org/docid/3df4be4fe.html (accessed on 11 October 2019).

**Ministry of Justice Statistics and Summaries**

Ministry of Justice (Japan), 2018 Report on the Status of Refugee Protection in Japan (Wagakuni ni okeru nanmin higo no jōkyō-tō). Available online: http://www.moj.go.jp/content/001290415.pdf (accessed on 12 October 2019).

Ministry of Justice (Japan), Cases Recognized as Refugees (Nanmin to shite nintei shita jirei-tō ni tsuite), in Statistics on Refugee Recognition in 2018 (Heisei 30-nen ni okeru nanmin ninteishasū-tō ni tsuite). Available online: http://www.moj.go.jp/content/001290417.pdf (accessed on 12 October 2019).

Ministry of Justice (Japan), Cases Recognized as Refugees (Nanmin to shite nintei shita jirei-tō ni tsuite), in Statistics on Refugee Recognition in 2017 (Heisei 29-nen ni okeru nanmin ninteishasū-tō ni tsuite). Available online: http://www.moj.go.jp/content/001257502.pdf (accessed on 12 October 2019).

Ministry of Justice (Japan), Cases Recognized as Refugees (Nanmin to shite nintei shita jirei-tō ni tsuite), in Statistics on Refugee Recognition in 2016 (Heisei 28-nen ni okeru nanmin ninteishasū-tō ni tsuite). Available online: http://www.moj.go.jp/content/001221349.pdf (accessed on 12 October 2019).

**Cases**

*Afghan v. Japan* (Prosecutor), Heisei 14 (2002) Wa (Criminal Case) No. 225, Japan: District Courts, 20 June 2002. Available online: https://www.refworld.org/cases,JPN_DC,428465274.html (accessed on 24 September 2019).

CJEU, case C-465/07, *Meki Elgafaji and Noor Elgafaji* v *Staatssecretaris van Justitie*, judgment of 17 February 2009. Available online: http://curia.europa.eu/juris/document/document.jsf?text=&docid=76788&pageIndex=0&doclang=EN&mode=lst&dir=&occ=first&part=1&cid=372339 (accessed on 24 September 2019).

*Turkish v. Japan* (Minister of Justice), Heisei 14 (2002) Gyo-U (Administrative Case) No.49, Japan: District Courts, 15 April 2004. Available online: https://www.refworld.org/cases,JPN_DC,4284b6a04.html (accessed on 24 September 2019).

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
