# Peer review of "Complementary Protection in Japan: To What Extent Does Japan Offer Effective International Protection for Those Who Fall Outside the 1951 Refugee Convention?"

_laws, 1951_

Round 1
Reviewer 1 Report
Complementary Protection in Japan
This study contributes to the growing body of scholarship on complementary protection for asylum seekers, shedding light on the poorly understood Special Permission to Stay (SPS) mechanism of complementary protection in Japan. Based on textual analysis of legal documents and case summaries published by the Japanese government, as well as a small number of elite interviews with Japanese immigration officials, the paper evaluates the merits and liabilities of SPS as a complementary protection mechanism, using the European Union’s Qualification Directive (QD) as an international benchmark. In the authors’ words, the goal of the study is to evaluate “the extent to which SPS provides effective international protection for those who are not recognized as refugees in Japan.”
Overall, this study is carefully researched and well executed, and makes a noteworthy contribution to the literature. However, there are several minor issues that require attention before the paper is ready for publication. Below is a detailed breakdown of comments and suggestions for each section of the paper.
Introduction
The rationale for this study is quite strong. The paper clearly addresses an important gap in the literature on complementary protection, by offering a focused cased study on Japan’s SPS program. Further, this paper’s timely publication has the potential to contribute to current policy debates in Japan, where the Diet is in the process of reforming the SPS program along the lines of the QD system in Europe. These are both compelling arguments, which are clear and well articulated in the paper.
Additionally, I would suggest adding a more explicit statement about the relative importance of SPS vis-à-vis Japan’s overall system of international protection for forced migrants. As the authors note in the introduction, Japan admits far more asylum seekers through SPS than it accepts as conventional refugees. In turn, SPS has more influence over the rights and protections afforded to asylum seekers in Japan than the 1951 Refugee Convention. In other words, SPS is the de facto arbiter of international protection for forced migrants in Japan. The authors would do well to highlight this point more clearly as a key part of the rationale for studying the SPS program in depth.
Methodology
The authors do a fine job explaining the types and sources of data collected and analyzed in this study. In particular, the case summaries published by the Japanese Ministry of Justice figure prominently as a main source of data throughout the analysis. This strikes me as a logical and fruitful approach. However, I do have some concerns about the limitations of this data. As the authors note, the Ministry chooses only a handful of successful SPS cases to summarize every year. We do not have any way of knowing how representative they are, or what they leave out. How does the government decide which cases to summarize and publish? Do these cases give an accurate picture of the wider universe of SPS cases? And what do we lose from not having any case summaries of asylees that were denied SPS status by the Ministry? These are not problems that the authors can easily correct, but I think it is important to acknowledge them more explicitly in the methodology section. In short, I would recommend a discussion of the limitations of the data and methodology, which highlights what we can learn from the available data, and what we cannot.
The second part of the methodology section focuses on why the European QD is an appropriate benchmark for evaluating the SPS system. I don’t have any issues with this approach in principle. However, the paper does not do a great job explaining why the QD is the ideal metric. What makes QD the gold standard? Why is it better than TPS in the United States, or Australia’s prohibition against refoulement? I have no doubt that the authors can provide a compelling answer, but I think it should be stated more clearly in the methodology section why the QD is the best framework for comparison.
Finally, the chart on page 6 could use a bit more explanatory detail. For example, what is the difference between “resettled refugees” and “recognized refugees”? (Is this third country resettlement vs. successful asylum applicants?) Moreover, where do these figures come from? The chart needs a legend that defines terms and identifies the data source.
Historical Background
This section is strong. However, I could not locate Table 1, which is referenced at least a couple of times in the text. This may be a simple clerical error, but it should be rectified prior to publication.
Evaluation of SPS
The thematic elements identified at the start of this section are logical and provide a coherent structure for the analysis. However, the introductory narrative at the beginning of this section (page 9, lines 290-299) is a bit awkward and abrupt. It could use a rewrite.
Overall, I found the analysis to be clear, well supported, and convincing. The authors do a good job providing evidence to back their claims about various aspects of the SPS program—including anecdotes from case summaries, interviewee comments, secondary source analyses, etc. I found the deductive logic used to draw inferences from some of the case summaries especially compelling. This is a creative and commendable use of the available data.
That being said, despite offering a thorough analysis of the content of SPS policy, the paper says relatively little about the lived experience of flesh-and-blood asylees with SPS status. While reading the paper, I kept coming back to some basic practical questions: What is it like to be a SPS designee in Japan? How is SPS protection actually exercised in practice? And how does that compare to the experience of conventional refugees? The paper offers a partial answer to these questions in sections 4.3 and 4.5, noting that SPS designees are denied access to the FWEAP course for refugees; they are not granted travel documents; and their residency status is granted for only 1 year at a time instead of 5 years for refugees. But I am still left wondering about what SPS status ultimately means for designees: How long do SPS designees typically stay in Japan? Is there a path to citizenship or permanent residency, or is SPS an inherently temporary status? What about children of SPS holders who are born in Japan—can they become permanent residents or citizens, despite the country’s jus sanguinis laws? Given the discretionary nature of SPS, do designees ever have their status revoked (or threatened) by the government? Are they subject to political attacks or societal discrimination?
Here, I am thinking about the TPS program in the United States, and the enormous gulf in rights and privileges between resettled refugees and TPS holders. Though the TPS program has offered a lifeline to many asylees in the U.S., it has also become a political football. Over the past several years, protection status for TPS holders has been threatened, cancelled, and ultimately extended at the last minute—repeatedly. While resettled refugees are granted permanent residency and citizenship if they choose it, TPS holders wait in perpetual limbo. It is an extremely stressful, uncertain, and ultimately precarious form of protection that does not come close to the standards enjoyed by conventional refugees.
While recognizing the value of the legal policy analysis in this paper, at some point I think the authors need to more explicitly address some of the practical implications of the SPS program for asylees who are denied refugee status and given complementary protection instead. It is up to the authors exactly where and how to do this, but it should be done.
Conclusion
This section is strong. It clearly highlights the main themes and key takeaways from the preceding analysis and points out critical areas for future research.
Reviewer 2 Report
interesting and well structured article
clear methodology, appropriate wrtitting tone, argumenation supported with bibliography
Referencing meeds standards hoewever some footnotes are in reality proper references and need to be included in the references section only
Author Response
Dear Sir/Madam
Good Day
We are extremely grateful for your valuable comments and minor revisions suggested, which will definitely enhance the quality of the paper.